# Autoregressive Frontier Expansion: Growing Trees with Graph Machine Learning

**Umer Gupta**
Independent Researcher
London, United Kingdom
umer.gupta152@gmail.com

**Saku Peltonen**
ETH Zürich
Zurich, Switzerland
speltonen@ethz.ch

**Martin Ritzert**
Center for Scalable Data Analytics and Artificial Intelligence (ScaDS.AI) Dresden/Leipzig
Universität Leipzig
Leipzig, Germany
martin.ritzert@uni-leipzig.de

## Abstract

Tree-like branching structures are common in nature, from botanical trees to neurons and respiratory trees. Their branching shape often reflects function, making structural modeling central to understanding how these systems work. Acquiring real-world 3D data via imaging can be expensive or infeasible, so realistic generative models are valuable for simulation and augmentation. Existing approaches either rely on hand-tuned, mechanistic procedures, or do not jointly generate both the tree topology and its 3D geometry. We propose a graph neural network architecture that generates trees through an iterative expansion process, simulating the biological growth of real trees. At each step, it grows the frontier by predicting whether each active branch should bifurcate or terminate. Experiments on botanical trees show that our method can learn the 3D branching structure across multiple trees.

## 1 Introduction

Tree-like (acyclic) branching structures regularly appear in nature: respiratory trees branch to increase surface area for gas exchange; botanical trees branch to improve light capture and resource distribution; and neurons branch to form connections with other neurons. Across biological systems, morphology – the anatomical shape – is often closely tied to physical constraints and functional requirements. For example, neuronal morphologies define the wiring patterns of the brain's circuitry and consequently, its function (Torben-Nielsen & De Schutter, 2014; Memelli et al., 2013; Lin et al., 2018).

Generating realistic simulations of branching structures is, hence, valuable for understanding such biological systems, especially where data capture is difficult. The brain, for example, contains billions of neurons with diverse morphologies (Markram et al., 2004) that need to be mapped to understand its function. This is resource intense (Schmitz et al., 2011), even for volumes as small as cubic millimetre of a mammalian brain (MICrONS Consortium, 2025). Beyond neuroscience, inpainting incomplete scans of trees directly leads to better estimation of their biomass. In both domains, branching structure is extracted from point clouds and data augmentation with realistic synthetic structures can improve this step.

Several classical approaches have been used to generate branching morphologies (Honda, 1971; Prusinkiewicz & Lindenmayer, 1990; Kanari et al., 2022), but work on learning-based methods remains limited. Recent efforts include MorphVAE (Laturnus & Berens, 2021) and MorphGrower (Yang et al., 2024) that use learning-based approaches to realise neuronal morphologies. MorphVAE decodes entire soma-to-tip walks *independently* (details in Section 1.1), failing to model the influence of previous branches on following ones (Yang et al., 2024). MorphGrower builds on the progressive nature of neuronal growth (Sarnat, 2023) and generates neuronal morphologies

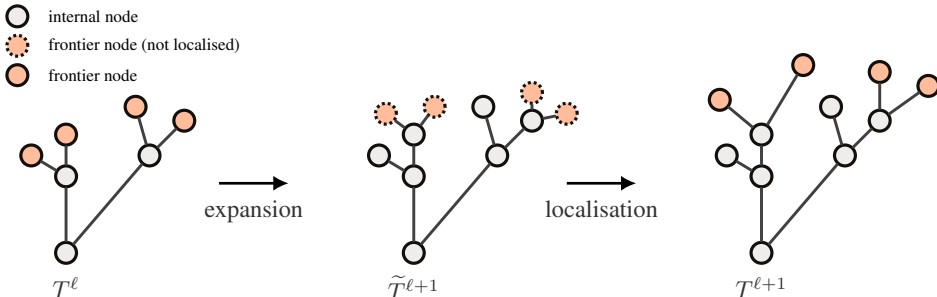

Figure 1: Overview of a generation step. Starting from a partial tree $T^\ell$, we (i) **expand** the current frontier (a subset of leaves) by deciding for each frontier node whether it branches or terminates, yielding a tree $\widetilde{T}^{\ell+1}$; then (ii) **localise** by predicting 3D parent-relative offsets for the previous frontier nodes, producing $T^{\ell+1}$. The process repeats until all frontier nodes terminate.

layer-by-layer, conditoning each layer on the previous one. However, the branching structure is *predefined* by a seed tree and only the geometry of points along the branches is modeled.

In this work, we propose an iterative procedure, *autoregressive frontier expansion*, which, similar to MorphGrower, mimics the growth patterns in nature, and conditions future branching on intermediate morphologies. However, unlike MorphGrower we do not restrict our branching to existing patterns in the data. Our method models both *topology*, "whether to branch or terminate", at each growth step as well as the *geometry* of each branch. Early results show that our method can learn both the branching pattern and 3D structure of multiple botanical trees.

## 1.1 RELATED WORK

**Growth-rule based approaches**    On the classical, non-learning side, growth-rule based methods use a predefined set of rules or probabilistic conditioning on priors to model the growth process of branching morphologies (Yang et al., 2024). Honda (1971) uses preset rules to model a tree as repeated bifurcations of straight segments with fixed branching angles and constant length contraction ratios. Topological Neuron Synthesis (TNS) (Kanari et al., 2022) simulates growth as a probabilistic procedure that generates neuronal trees conditioned on a topological description of their branching structure. The conditioning signal is given by persistence barcodes (Carlsson, 2009), formalized for complex branching morphologies by Kanari et al. (2018) as the Topological Morphology Descriptor (TMD).

**Learned tree generation**    Existing learned methods for generating neuronal trees include MorphVAE (Laturnus & Berens, 2021), which trains a sequence-to-sequence variational autoencoder on soma-to-tip walks. New trees are sampled by decoding a set of such walks in 3D, clustering nearby points to merge nodes, and connecting consecutive points along each walk to form edges. MorphGrower (Yang et al., 2024) grows neuronal trees iteratively, expanding the current tree by generating sibling branch pairs simultaneously, which improves plausibility. However, the branching structure is fixed to that of a seed tree, and only new geometry is generated. Bergmeister et al. (2023) showed that iterative expansion can be used to generate all kinds of sparse graphs including trees, even though the paper did only focus on the pure graph structure, while predicting 3D positions for each node are a core component in our work.

More broadly, TreeGen (Kollovieh et al., 2025) is a learned generative model for rooted hierarchies that works by iteratively denoising a probabilistic parent-assignment representation. Tree generation also appears as a subroutine in general learnable graph generation algorithms. For example, Jin et al. (2018) generate molecules by generating their junction tree that serves as a backbone for the graph.

## 2  METHODOLOGY

A tree $T$ is a tuple $(V, E, P)$ where $V = \{1, \ldots, N\}$ are nodes, $E \subset V \times V$ are edges, and $P \in \mathbb{R}^{|V| \times 3}$ are node coordinates (root-centered). We assume that node 1 is always the root. Each non-root node $v \neq 1$ has a unique parent $\pi(v)$. Let $\mathcal{L}(T) \subseteq V$ denote the set of leaves of the tree. By $X(i)$ we denote the $i$-th row of $X$.

For modeling (botanical) trees we restrict ourself to the overarching branching geometry and thus only consider branching points (bifurcations) and leaves (terminations/tips), see Appendix A. We can do so since most information is encoded in the "orientation and length of new branches" for both botanical trees (Honda, 1971) and neurons (Kanari et al., 2018).

### 2.1  TREE EXPANSION

We generate trees iteratively as a sequence $\{T^0, \ldots, T^L\}$. A partial tree at level $\ell$ is $T^l = (V^\ell, E^\ell, P^\ell)$. Our generation operates on an *active frontier* $\mathcal{A}^\ell \subseteq \mathrm{Leaves}(T^\ell)$, i.e., the leaves created in the previous step. Each iteration consists of two distinct operations: (1) frontier expansion and (2) localisation. See Fig. 1 for an overview and Algorithm 4 for a pseudocode.

**Frontier expansion.**  Given $T^\ell$, we predict a binary expansion state $\Gamma^\ell(v) \in \{0, 1\}$ for each $v \in \mathcal{A}^\ell$ and attach two children $(v_L, v_R)$ to every $v$ with $\Gamma^\ell(v) = 1$. Together with the new edges $(v, v_L)$ and $(v, v_R)$ this defines the intermediary expanded tree $\tilde{T}^{\ell+1}$.

**Localisation.**  In the intermediary tree $\tilde{T}^{\ell+1}$ the coordinates of the new leaves $v \in \mathcal{A}^{\ell+1}$ are not yet specified. For such nodes $v \in \mathcal{A}^{\ell+1}$ we get the coordinates $P^{\ell+1}(v)$ and thus the next partial tree $T^{\ell+1}$ by predicting parent-relative offsets $C^{\ell+1}(v)$:

$$P^{\ell+1}(v) = P^\ell(\pi(v)) + C^{\ell+1}(v), \tag{1}$$

Offsets $C^{\ell+1}(v)$ are predicted conditioned on the intermediary tree $\tilde{T}^{\ell+1}$. For all nodes not in the frontier $\mathcal{A}^{\ell+1}$, the position does not change, so $P^{\ell+1}(u) = P^\ell(u)$ for $u \notin \mathcal{A}^{\ell+1}$.

Instead of alternating between expansion and localisation, in our implementation we jointly predict the positions $P^{\ell+1}$ and the expansion states $\Gamma^{\ell+1}$ for the next expansion step of all leaves in the frontier $\mathcal{A}^{\ell+1}$ conditioned on $\tilde{T}^{\ell+1}$. In addition, the prediction of the positions and expansion steps follows a $\sigma$-conditioned denoising diffusion model (see Appendix B), i.e. we use several diffusion steps to find the best-fitting position and expansion state for each node in $\mathcal{A}^{\ell+1}$ before fixing it for future rounds.

### 2.2  MODELING AND TRAINING

**Sampling.**  We initialize $T^0 = (\{r\}, \emptyset, P^0(r) = \mathbf{0})$ and assume the root *has* to expand: $\Gamma^0(r) = 1$. That is, $\tilde{T}^1$ is defined and $\mathcal{A}^1 := \{r_R, r_L\}$. For $\ell \geq 1$, we can repeatedly sample positions $P^\ell$ and expansion states $\Gamma^\ell$ conditioned on $\tilde{T}^\ell$, defining the tree $T^\ell$ as well as $\tilde{T}^{\ell+1}$ with the new frontier $\mathcal{A}^{\ell+1}$ according to $\Gamma^\ell$. We continue until $\Gamma^\ell = \mathbf{0}$, i.e. no nodes will expand. Note that for our initial study we stop generation when $|T^\ell| = |T^\star|$ where $T^\star$ is the ground-truth tree we condition on.

**Equivariant GNN.**  We want our model to be equivariant wrt. rotations around the z-axis, but not around all axes since trees are (typically) growing upwards and neurons also have a primary axis, namely perpendicular to the cortical surface (Cajal, 1911). We thus use a novel SO(2)-EGNN model (see Appendix C) inspired by Satorras et al. (2021) who mainly looked at full E(n) equivariance.

**Coarsening sequence.**  The model is trained on ground truth samples (expansion state and position) at each level created using a deterministic backward process: at level $\ell$ we keep all nodes that are at most $\ell$ hops from the root. We define $\mathcal{A}^\ell$ to contain all nodes with $d(r, u) = \ell$, i.e. the deepest leaves. This leaves us with only the root at $\ell = 0$.

Table 1: Results from reconstruction. For each tree, ground truth (GT) and predicted values are reported on separate rows for each statistic.

| Tree | | $|V|$ | MBPL | MASB | Height | Radial Span | Box Diagonal | Chamfer Distance | TED |
|------|------|------|------|------|--------|-------------|--------------|-------------------|-----|
| Tree 1 | *GT* | 149 | 0.34 | 64.42 | 7.65 | 4.32 | 9.33 | 0.08 | 0.01 |
| | *Pred* | 145 | 0.36 | 64.41 | 7.77 | 4.46 | 9.55 | | |
| Tree 2 | *GT* | 235 | 0.34 | 69.31 | 3.55 | 6.12 | 8.38 | 0.23 | 0.02 |
| | *Pred* | 235 | 0.32 | 72.15 | 3.59 | 6.37 | 8.43 | | |
| Tree 3 | *GT* | 327 | 0.36 | 81.50 | 6.37 | 9.46 | 12.76 | 0.39 | 0.03 |
| | *Pred* | 315 | 0.36 | 80.82 | 6.35 | 9.04 | 12.50 | | |

**Global conditioning.** To guide generation in early steps, when trees look similar, we condition each diffusion step on a compact topological descriptor of the complete tree computed using 0D persistent homology (Edelsbrunner et al., 2008) under simple filtrations – root path length (superlevel), height (sublevel), and radial distance (sublevel) (see Appendix D). At present, to guide expansion, we additionally pass the current size of the partial graph as a fraction of the size of the target graph.

## 3 RESULTS

In this section, we present the results of our procedure on reconstruction of a dataset of 3 botanical tree crowns of varying sizes – 149, 235 and 327 nodes respectively – and a maximum depth of 10 from the root. The results are depicted in Fig. 2 showing the ground-truth and predicted skeletons. One can see that the overall structure is captured well with minor mistakes in the exact positions of branches.

For a quantitative evaluation, we compute height, diameter, bounding-box diagonal, as well as the means of branch path length and angle between sibling branches in Table 1. We additionally turn the trees into pointclouds by sampling at 1cm intervals and compute the L1 Chamfer Distance (Nguyen et al., 2021). For structural similarity we compute the tree edit distance. Detailed metric definitions can be found in Appendix E and histograms of branch lenght and bifurcation angles are in Fig. 2. We observe generally matching statistics for all three trees. The relatively high Chamfer Distance for Trees 2 and 3 can be explained by small early errors in the generation which lead to entire branches becoming shifted. This massively increases the Chamfer Distance even though the overall structure is practically identical. TED is extremely small for all three trees, indicating that we indeed learned to produce the correct structure. Further, both histograms in Fig. 2 and statistics in Table 1 are very close for all three trees, supporting faithful generation.

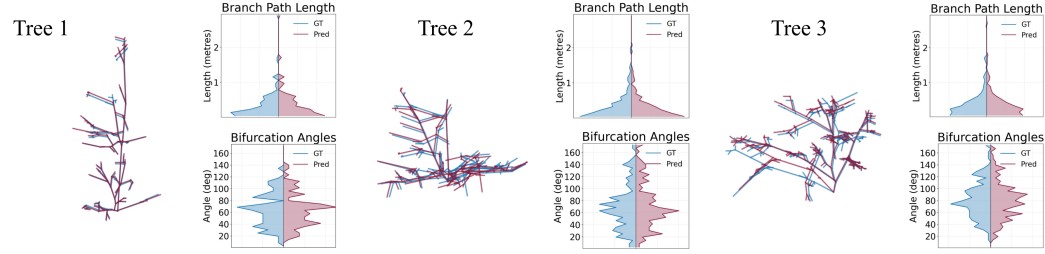

Figure 2: Generated trees (red) and ground truth (blue). The histograms show the distribution of branch lengths and bifurcation angles across the 3 trees. Both statistics and structure match the GT.

## 4 DISCUSSION

Learning-based methods that both generate the topology and geometry of branching tree structures like botanical trees and neuronal anatomy are few with most graph generative methods focusing on molecules instead. We successfully learn branching structures of botanical trees up to 300 nodes,

setting the foundation for a learning-based method for branching structures capable of generating new, realistic and diverse trees from the underlying data distribution.

**Limitations and future work.** At present, we validate the methodology on a small subset of trees and use a supervision target $C^\ell$ defined in the global frame to simplify evaluation. Extending the objective to be fully SO(2)-equivariant is a natural next step. We expect this inductive bias to be beneficial when learning to generate novel trees, particularly in low-data regimes.

During generation, we currently assume that due to sparsity, generated structures do not intersect. In the future, we plan to train the model to avoid collisions and also take into account the competition for light or space, i.e. surrounding trees or neurons.

For evaluation, we would like to have a similarity metric that takes both the structure and positions into account. It should also be less strict against (small) shifts of individual branches as long as the local structure is kept, possibly following Wang et al. (2023) or building on the Gromov-Wasserstein distance.

We believe that the topological morphological guidance signal supports generation beyond disambiguating early steps (Appendix D). Learning an underlying latent distribution on these descriptors to sample and conditon on is an active area of exploration for us.

## 5    ACKNOWLEDGEMENTS

The authors acknowledge the financial support by the Federal Ministry of Research, Technology and Space of Germany and by Sächsische Staatsministerium für Wissenschaft, Kultur und Tourismus in the programme Center of Excellence for AI-research "Center for Scalable Data Analytics and Artificial Intelligence Dresden/Leipzig", project identification number: ScaDS.AI

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

# A   TREE REPRESENTATION

In our experiments, we assume that all trees are binary trees which means that at every point only a single branch may branch off. This binary condition simplifies the decision space for the learned expansion target. While this constraint holds true for most dedicious trees, others such as the fir tree typically have two branches on opposite sites of the tree. In such cases, we introduce an additional node at the intersection to turn a trifucation into two bifurcations. Note that our model and generation procedure can easily be extended to work with higher-order splits by making $\Gamma$ categorical and assuming some sensible upper limit on the maximum degree of a node.

# B   DIFFUSION

At each expansion step, the newly created frontier $\mathcal{A}^{\ell+1}$ contains leaves whose coordinates and *next-step* expansion state must be generated. We parametrise this as a denoising diffusion process on the *frontier state*, $X^{\ell+1} = [C^{\ell+1}, \Gamma^{\ell+1}] \in \mathbb{R}^{|\mathcal{A}^{\ell+1}| \times 4}$, conditioned on the full intermediate expanded tree $\tilde{T}^{\ell+1}$. For diffusion, we treat $\Gamma$ as continuous.

In this section, we drop the tree expansion level $\ell + 1$ from the superscript to make the notation easier to read. In the subscript we now write the noise level in the diffusion process which we introduce next, so $P_t^{\ell+1}(v)$ is the position of node $v$ in round $\ell + 1$ while being at noise level $t$ during the diffusion process.

**Forward noising.**   We use a continuous noise level $t > 0$ and apply Gaussian corruption in frontier-state space:

$$X_t = X_0 + t\,\varepsilon, \qquad \varepsilon \sim \mathcal{N}(0, I_4) \tag{2}$$

where $I_4 \in \mathbb{R}^{4 \times 4}$ is the 4-dimensional identity. In training we sample $t$ per graph from a log-normal distribution. In general, we are sampling $X_0^{\ell+1}$ for training and are adding noise to it to get $X_t^{\ell+1}$. Thus, each training example consists of a tree $T$ sampled at some level $\ell + 1$ together with a noise value $t$ and we aim to predict $X_0^{\ell+1}$ from $X_t^{\ell+1}$.

**Denoiser and Inputs.**   Our denoiser is the SO(2)-EGNN model which consumes absolute coordinates. We therefore construct $P_t$ by overwriting only the frontier node coordinates using the noised offsets:

$$P_t(v) = \begin{cases} P_0(v), & v \notin \mathcal{A}, \\ P_0(\pi(v)) + C_t(v), & v \in \mathcal{A}. \end{cases} \tag{3}$$

All non-frontier nodes remain clean and provide geometric context.

Let $H$ denote deterministic graph inputs: node features and constructed geometric edge features. We provide $\log t$ as a broadcast node feature and provide the noisy expansion state $\Gamma_t$ as an additional node feature on frontier nodes (and $0$ elsewhere). Then

$$D_\theta : \ (P_t, H, \Gamma_t, t) \mapsto \widehat{X}_0, \tag{4}$$

and to give frontier state predictions $\widehat{X}_0 = [\widehat{C}_0, \widehat{\Gamma}_0]$.

**Training Objective.**   We train an $X_0$-predictor with an MSE loss on the fontier nodes only. Let $\theta$ denote the parameters of the SO(2)-EGNN.

$$\mathcal{L}(\theta) = \mathbb{E}_{t,\varepsilon}\left[ \sum_{i \in \mathcal{A}} \|\widehat{C}_0(i) - C_0(i)\|_2^2 + \sum_{i \in \mathcal{A}} (\widehat{\Gamma}_0(i) - \Gamma_0(i))^2 \right]. \tag{5}$$

**Deterministic Sampling**   At inference, we sample by starting from Gaussian noise at $t_{\max}$ and denoising along a decreasing schedule $t_0 > \cdots > t_K > t_{K+1} = 0$. Given current $X_{t_k}$, we predict $\widehat{X}_0 = D_\theta(P_{t_k}, H, \Gamma_{t_k}, t_k)$ and form

$$\widehat{\varepsilon} = \frac{X_{t_k} - \widehat{X}_0}{t_k}, \qquad X_{t_{k+1}} = \widehat{X}_0 + t_{k+1}\widehat{\varepsilon}. \tag{6}$$

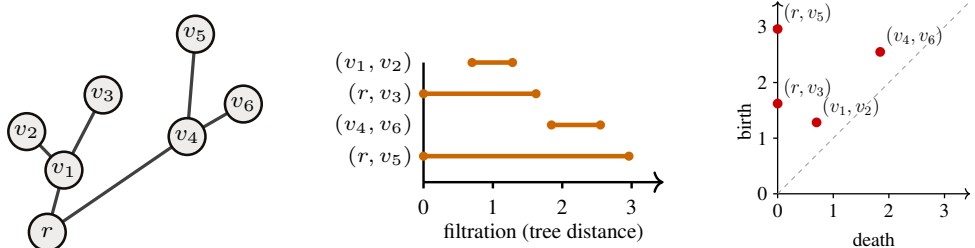

Figure 3: **(Left)** A tree with root $r$. The nodes are embedded in $\mathbb{R}^2$. Their coordinates induce Euclidean edge lengths, which defines a distance-to-root filtration. **(Middle)** A barcode showing the resulting $H_0$ persistence under this filtration, with the elder rule deciding which branch survives each merge. The bars are annotated by $(v_d, v_b)$, the death and birth nodes respectively. Note that the barcode itself is unlabeled. **(Right)** The corresponding persistence diagram. Points close to the diagonal represent short-lived features, while points further away represent more persistent features.

After the final step ($t_{K+1} = 0$), we use $\widehat{C}_0$ to place the frontier nodes and threshold $\widehat{\Gamma}_0$ to obtain binary expansion decisions for the next iteration.

The diffusion design space has not been explored in this work. In following works, we want to include improvements proposed by Karras et al. (2022) for better training and sampling dynamics.

## C    SO(2)-EGNN

The original EGNN achieves E(n) equivariance by using only pairwise squared distances as geometric input to edge messages, deliberately avoiding directional information that would break rotational symmetry (Satorras et al., 2021). For structures with a preferred axis, we instead decompose displacements into axis-parallel and perpendicular components.

For each directed edge $(i \rightarrow j)$ with coordinates $\mathbf{x}_i, \mathbf{x}_j$, we compute the relative displacement $\mathbf{r}_{ij} = \mathbf{x}_j - \mathbf{x}_i$ and decompose it w.r.t. a fixed rotation axis $\hat{\mathbf{u}}$ (default $z$-axis). We form the axial component $d_{ij}^{\parallel} = \mathbf{r}_{ij}^{\top}\hat{\mathbf{u}}$ and the perpendicular residual $\mathbf{r}_{ij}^{\perp} = \mathbf{r}_{ij} - d_{ij}^{\parallel}\hat{\mathbf{u}}$, then take $\rho_{ij} = \|\mathbf{r}_{ij}^{\perp}\|_2$. The pair $(\rho_{ij}, d_{ij}^{\parallel})$ is invariant to rotations around $\hat{\mathbf{u}}$. We enrich these scalars with angular information in the local parent-centric frame in the plane perpendicular to $\hat{\mathbf{u}}$ (default $xy$-plane) – $\cos\psi, \sin\psi$ – as well as $\cos\phi$ measuring the tilt of the outgoing branch relative to $\hat{\mathbf{u}}$. Parent-centric frames are formed based on the orientation of the parent branch, i.e., (*grandparent* → *parent*) edge, projected onto the perpendicular plane.

## D    TOPOLOGICAL MORPHOLOGICAL CONDITIONING

To generate trees with a desired structural profile, we guide sampling with a compact topological descriptor of global morphology. This is necessary because generation starts from a single node with position $\mathbf{0}$, which contains no information about the target morphology. Below we describe the relevant topological background in the context of tree morphologies.

**Filtrations.** Let $T = (V, E, P)$ be a tree with root $r$. Each node $v \in V$ has coordinates $P(v) \in \mathbb{R}^3$. A *filtration function* is a map $f : V \rightarrow \mathbb{R}$. Given $f$, we can define a filtration of $T$ as a nested sequence of induced subgraphs $\{T_\alpha\}_{\alpha \in \mathbb{R}}$ where $T_\alpha$ contains all nodes $v$ with $f(v) \leq \alpha$ (sublevel set filtration) or $f(v) \geq \alpha$ (superlevel set filtration).

**Barcode and elder rule.** In this work we only use 0-dimensional persistent homology ($\mathrm{PH}_0$), which tracks how connected components appear and merge along the filtration. A component is born at the filtration value $\alpha$ when a new connected component appears in $T_\alpha$. Two components merge at filtration value $\alpha$ when they become connected in $T_\alpha$. When two components merge, we apply the *elder rule*: the older component (smaller birth time) persists, and the younger component dies.

The multiset of intervals $\{\{[b_i, d_i)\}\}_i$ where $b_i$ and $d_i$ are the birth and death times of component $i$ is called the *persistence barcode* of $T$ under filtration $f$. See Figure 3 for an example.

**Persistence diagram.**   A barcode can equivalently be represented as a multiset of points in $\mathbb{R}^2$, called the *persistence diagram*, where each point $(b_i, d_i)$ corresponds to a bar $[b_i, d_i]$ in the barcode. Points close to the diagonal represent short-lived features, while points further away represent more persistent features.

**Filtration choices used in this work.**   We compute $\text{PH}_0$ barcodes of the tree under the following filtrations:

- Sublevel set filtrations based on height $z$ and radial distance $\sqrt{x^2 + y^2}$: can highlight non-monotonic geometric variation, highlighting persistent "valleys" caused by branches that dip downward or curve back toward the axial trunk (Beers & Leygonie, 2025).
- Superlevel set filtration based on path distance from the root (see Figure 3): captures the branching structure, yielding a compact summary of the global branching allocation across depth – *where* and *how much* branching occurs.

**Vectorisation.**   To provide this morphological information to the network with a fixed dimension, we convert persistence diagrams into persistence images. A *persistence image* (PI) (Adams et al., 2017) is a fixed-dimensional vector representation of a persistence diagram obtained by placing Gaussian kernels at each point and discretizing the resulting density on a grid. The PIs for all three filtrations are flattened and concatenated into a single conditioning vector, which is embedded and broadcast to nodes as a graph-level condition. In practice, this reduces early-step ambiguity and encourages trees whose branching structure and geometric trends match the target morphology.

## E   METRICS

This section contains definitions of the metrics used in Table 1. We use $T = (V, E, P)$. Let $x_v, y_v, z_v = P(v)$ denote the position of a node $v \in V$.

**MBPL**   Mean branch path length:

$$\text{MBPL}(T) = \frac{1}{|E|} \sum_{(u,v) \in E} \big\| P(v) - P(u) \big\|_2$$

**MASB**   Mean angle between sibling branches: Average over the angle in degrees between the two children of each inner node

$$\text{MASB}(T) \;=\; \frac{1}{|\mathcal{I}|} \sum_{u \in \mathcal{I}} \deg\big( P(c_1(u)) - P(u),\; P(c_2(u)) - P(u) \big)$$

where $c_1(u)$ and $c_2(u)$ denote the two children of the inner node $u$ and $\mathcal{I}$ denotes the set of inner nodes of $T$.

**Height**   Height of the tree crown in meter

$$\text{Height}(T) = \max_{v \in V} z_v$$

**Radial Span**   Project the tree into the XY-plane, then take the longest distance between any two points in the projection in meter

$$\text{Diameter}(T) = \max_{u,v \in V} \big\| (x_u, y_u) - (x_v, y_v) \big\|_2$$

**Box Diagonal**   The diagonal of an axis-aligned bounding box in meters.

**Chamfer Distance**   Pointcloud-based metric, each tree is transformed into a pointcloud by sampling it at 1cm intervals

$$\text{CD}(T_1, T_2) = \frac{1}{|T_1|} \sum_{u \in V_1} \min_{v \in V_2} \| P_1(u) - P_2(v) \|_2^2 \;+\; \frac{1}{|T_2|} \sum_{u \in V_2} \min_{v \in V_1} \| P_2(u) - P_1(v) \|_2^2.$$

**TED** Normalized tree edit distance. TED evaluates structural similarity between trees by measuring how many edit operations are necessary to turn one tree into another. We use unit cost for insertion and deletion and assume that there are no label names since we only care about the structural similarity. Since the raw TED scales with the number of nodes, we normalize it with the maximum possible TED, in our case $|T_1| + |T_2|$. The minimum possible TED is $\big||T_1| - |T_2|\big|$ which is achieved on Tree 1.

Since TED relies on an ordering on the children, we order them by the size of the induced subtree (recursively in the case of a tie).

## F PSEUDOCODE

---
**Algorithm 1:** Tree generation procedure

---
1   $T^0 = (\{r\}, \emptyset, P^0(r) = \mathbf{0})$          *// Initialize tree with root node $r$ at the origin*
2   $\mathcal{A}^0 = \{r\}$          *// Initialize frontier with the root node*
3   $\ell = 0$
4   **while** $\mathcal{A}^\ell \neq \emptyset$ **:**
5      Predict expansion using diffusion$^\dagger$: $\Gamma^\ell \in \{0,1\}^{|\mathcal{A}^\ell|}$
6      $V^{\ell+1} = V^\ell; \; E^{\ell+1} = E^\ell; \; \mathcal{A}^{\ell+1} = \emptyset$
7      **foreach** $v \in \mathcal{A}^\ell$ *with* $\Gamma^\ell(v) = 1$ **:**
8         $V^{\ell+1} = V^{\ell+1} \cup \{v_L, v_R\}$
9         $E^{\ell+1} = E^{\ell+1} \cup \{(v, v_L), (v, v_R)\}$
10        $\mathcal{A}^{\ell+1} = \mathcal{A}^{\ell+1} \cup \{v_L, v_R\}$
11     Predict offsets using diffusion: $C^{\ell+1} \in \mathbb{R}^{n' \times 3}$        *// $n' = |V^{\ell+1}| - |V^\ell|$*
12     $P^{\ell+1} = P^\ell$
13     **foreach** $v \in V^{\ell+1} \setminus V^\ell$ **:**
14        $P^{\ell+1}(v) = P^\ell(\pi(v)) + C^{\ell+1}(v)$        *// $\pi(v)$ is the parent of $v$*
15     $\ell = \ell + 1$
16   **return** $T^\ell$

---

Figure 4: Autoregressive tree generation. In each iteration, the current frontier $\mathcal{A}^\ell$ is expanded according to the predicted expansion state $\Gamma^\ell$. New nodes are added to the tree and their positions are defined by the predicted offsets $C^{\ell+1}$ relative to their parent. The process continues until no nodes in the frontier are set to expand. ($\dagger$) For $\ell = 0$, we set $\Gamma^0(r) = 1$ to force the first expansion.

# G  STEP-WISE VISUALISATION OF A RECONSTRUCTED MORPHOLOGY

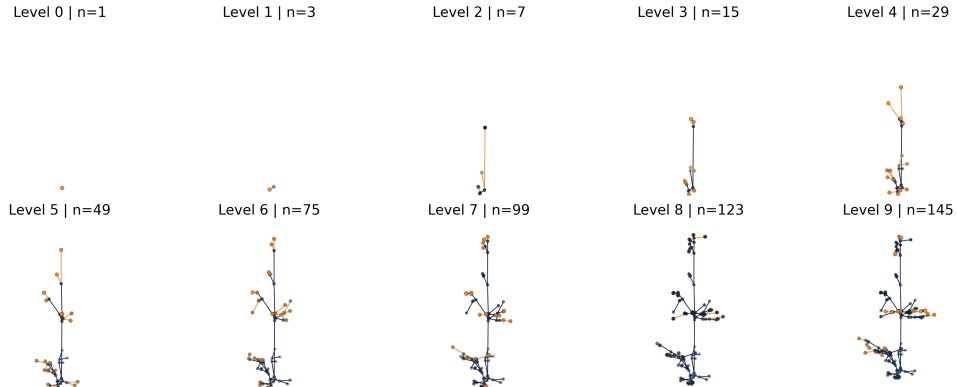

Figure 5: Step-wise generation of Tree 1. New nodes and edges in orange, existing nodes and edges in blue.

# H  MORE VISUALISATIONS OF GT VS RECONSTRUCTIONS

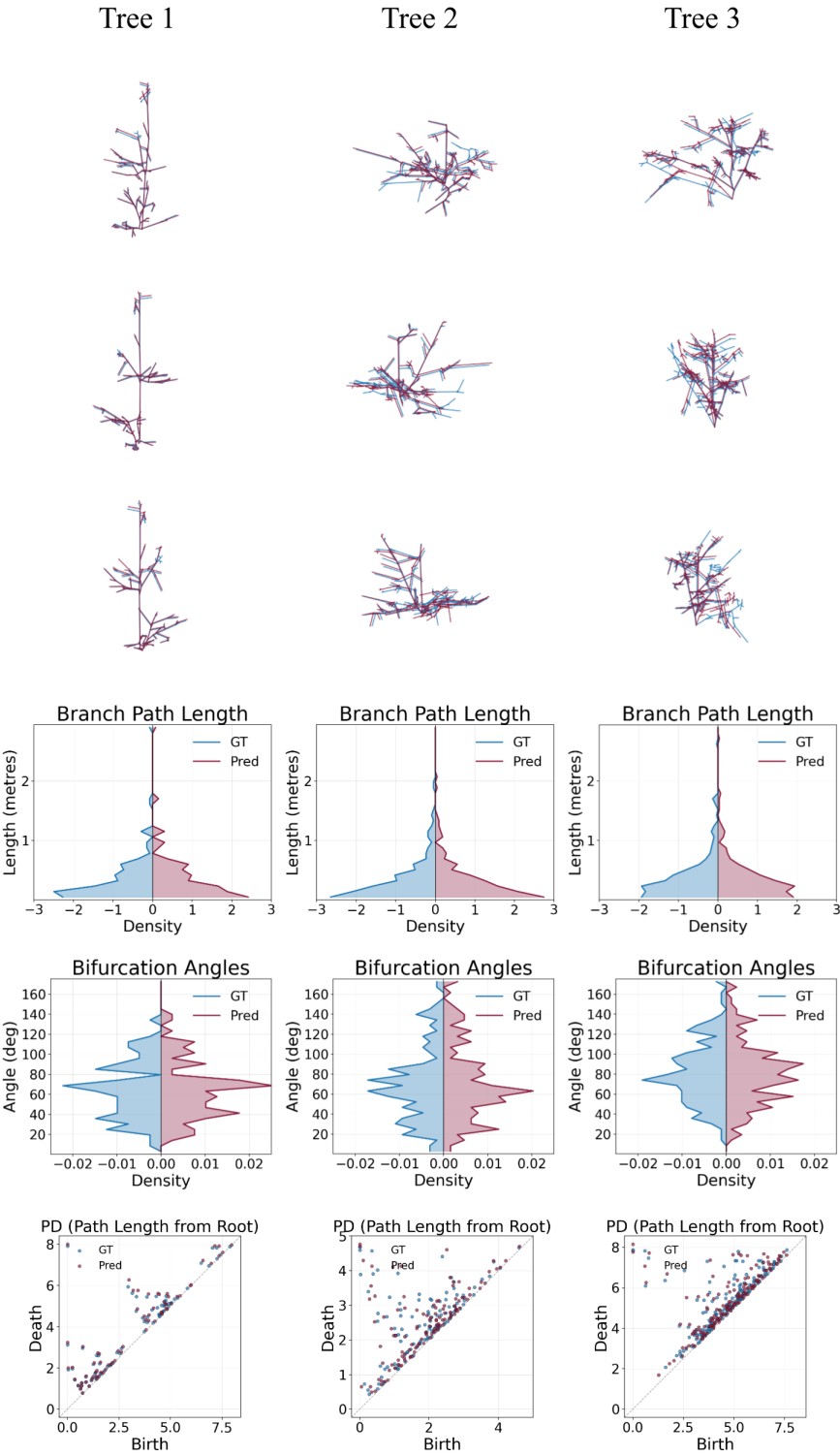

Figure 6: More complete visualization of the three trees including histograms and the persistence diagrams from Fig. 3.

