# OpenReview forum: "Autoregressive Frontier Expansion: Growing Trees with Graph Machine Learning"
_ICLR.cc/2026/Workshop/GRaM — ICLR 2026 Workshop GRaM Poster_

### Official Review · Reviewer_dR1T · 2026-02-13
**A clever, biologically-inspired approach to 3D branching structures**

**Rating:** 6
**Confidence:** 4

**Review:**

This paper introduces an autoregressive framework for growing tree-like structures (such as neurons or botanical trees) by jointly generating topology and 3D geometry through a "frontier expansion" logic. Unlike standard generative models that predict a global graph structure, this approach grows the tree node-by-node, which mirrors natural biological growth and allows for much better local structural control. The step-wise visualizations of the reconstruction process are compelling and suggest that the model effectively learns the recursive branching rules of 3D data. However, the paper is noticeably thin on methodological details regarding physical constraints; it does not address how the model handles branch self-collisions or competition for space, which are fundamental in mechanistic tree modeling.  While the lack of physical guardrails is a weakness, the core generative strategy is highly original and a strong fit for the workshop’s focus on relational geometry.

**Pmlr Suitability:**

NA

---

### Official Review · Reviewer_NpLK · 2026-02-20

**Rating:** 6
**Confidence:** 4

**Review:**

The paper proposes a method called *autoregressive frontier expansion*, which aims to use machine learning techniques to generate trees. In particular, the authors claim that while pre-existing approaches rely on mechanistic procedures, their method, which uses a graph neural network, is more flexible in the sense that it performs an iterative expansion process.

**Strengths**
- The abstract and introduction are clear, well-structured, and accessible to non-experts.
- The contributions are clearly stated, and the work addresses research questions in an under-explored area, proposing novel solutions.
- The paper is overall well-structured with a sufficient amount of clarifying images and tables.

**Weaknesses/Questions**
- Although the contributions are clear, the motivations aren't. When reading through the methodology section, I couldn't easily find an answer to "why?". In particular: (1) Why did you choose this specific technique in the first place? (2) How does this method reach your goals and answer your research questions better than other methods?
- The methodology section is quite technical and discursive. To avoid losing the sense of the general picture, it would be useful to include an algorithm block or a simplified schema to give an intuitive overview of the whole procedure.
- What research questions are you aiming to answer through your experiments, and do the results show what you claim in the rest of the paper? In the introduction and abstract, you claim that your method provides advantages over previous methods (MorphVAE, MorphGrower) by being able to model both the topology and geometry of each branch. Therefore, I would expect the experiments to showcase these advantages in some way. Instead, the experiments only solve a reconstruction problem, comparing the generated trees to the original ones, which doesn't fully demonstrate the flexibility of your model. It would be highly beneficial to include a comparison with the already existing techniques.

**Additional comments**
- I would recommend making "Related Work" a proper section instead of a subsection of the introduction.
- This is more of a personal curiosity: in the introduction, you claim that "Generating realistic simulations of branching structures is, hence, valuable for understanding such biological systems ...". In what specific ways is it helpful?

**Pmlr Suitability:**

NA

---

### Official Review · Reviewer_McV9 · 2026-02-23

**Rating:** 6
**Confidence:** 2

**Review:**

Summary
This work proposes a learning-based framework for 3D tree generation. The method trains a network to predict (i) the 3D position (as parent-relative offsets) and (ii) the expansion state (branch vs terminate) of frontier leaves in the current partial tree. By repeatedly applying these predictions starting from a single root node, the model grows the tree iteratively. Experiments on three separate tree datasets show reasonable reconstruction quality.

Strengths
- Framing tree generation as a learning problem is well-motivated and a promising direction.
- Using an equivariant network (SO(2) in particular) is a sensible inductive bias for trees with a preferred growth axis.
- The experimental results demonstrate reasonable reconstruction fidelity on the evaluated datasets.

Weakness
- The current evaluation focuses primarily on reconstruction of a small set of trees. It would be more compelling to demonstrate scaling to broader datasets and/or generalization to unseen trees.
- Relatedly, evaluating whether the model can generate novel trees (not reconstructions) that preserve similar structural patterns (e.g., distributional statistics/topological summaries) would strengthen the generative claims.

For a tiny paper/workshop track, this is a well-motivated and technically sound step toward learning-based generation of branching structures that jointly model topology and geometry. While the current evaluation is small-scale and mostly reconstructive, the method is promising and clearly articulated, and it provides a solid foundation for larger-scale generative experiments.

**Pmlr Suitability:**

NA

---

### Meta-Review · Area_Chair_f95i · 2026-02-26

**Decision:**

Accept

**Metareview:**

The reviewers agree that this contribution is original and of interest to the workshop. Authors are encouraged to improve clarity and evaluation. I am happy to recommend acceptance.

**Relevance To Proceedings:**

Tiny paper — does not apply

**Relevance To Workshop:**

Yes — suitable for GRaM

---

### Decision · Program_Chairs · 2026-03-02

Accept (Poster)